# Wavelet Score-Based Generative Modeling

**Florentin Guth**
Computer Science Department,
ENS, CNRS, PSL University

**Simon Coste**
Computer Science Department,
ENS, CNRS, PSL University

**Valentin De Bortoli**
Computer Science Department,
ENS, CNRS, PSL University

**Stéphane Mallat**
Collège de France, Paris, France
Flatiron Institute, New York, USA

## Abstract

Score-based generative models (SGMs) synthesize new data samples from Gaussian white noise by running a time-reversed Stochastic Differential Equation (SDE) whose drift coefficient depends on some probabilistic score. The discretization of such SDEs typically requires a large number of time steps and hence a high computational cost. This is because of ill-conditioning properties of the score that we analyze mathematically. Previous approaches have relied on multiscale generation to considerably accelerate SGMs. We explain how this acceleration results from an implicit factorization of the data distribution into a product of conditional probabilities of wavelet coefficients across scales. The resulting Wavelet Score-based Generative Model (WSGM) synthesizes wavelet coefficients with the same number of time steps at all scales, and its time complexity therefore grows linearly with the image size. This is proved mathematically for Gaussian distributions, and shown numerically for physical processes at phase transition and natural image datasets.

## 1 Introduction

Score-based Generative Models (SGMs) have obtained remarkable results to learn and sample probability distributions of image and audio signals [44, 3, 24, 38, 39, 6]. They proceed as follows: the data distribution is mapped to a Gaussian white distribution by evolving along a Stochastic Differential Equation (SDE), which progressively adds noise to the data. The generation is implemented using the time-reversed SDE, which transforms a Gaussian white noise into a data sample. At each time step, it pushes samples along the gradient of the log probability, also called *score function*. This score is estimated by leveraging tools from score-matching and deep neural networks [13, 47]. At sampling time, the computational complexity is therefore proportional to the number of time steps, i.e., the number of forward network evaluations. Early SGMs in [44, 46, 11] used thousands of time steps, and hence had a limited applicability.

Diffusion models map a Gaussian white distribution into a highly complex data distribution. We thus expect that this process will require a large number of time steps. It then comes as a surprise that recent approaches have drastically reduced this time complexity. This is achieved by optimizing the discretization schedule or by modifying the original SGM formulation [18, 17, 27, 53, 42, 37, 43, 23, 11, 29, 41, 51]. High-quality score-based generative models have also been improved by cascading multiscale image generations [40, 12, 6] or with subspace decompositions [16]. We make explicit the reason of this improvement, which provably accelerates the sampling of SGMs.

A key idea is that typical high-dimensional probability distributions coming from physics or natural images have complex multiscale properties. They can be simplified by factorizing them as a product of conditional probabilities of normalized wavelet coefficients across scales, as shown in [33].

36th Conference on Neural Information Processing Systems (NeurIPS 2022).

These conditional probabilities are more similar to Gaussian white noise than the original image distribution, and can thus be sampled more efficiently. On the physics side, this observation is rooted in the renormalization group decomposition in statistical physics [49], and has been used to estimate physical energies from data [33]. In image processing, it relies on statistical observations of wavelet coefficient properties [48]. A Wavelet Score-based Generative Model (WSGM) generates normalized wavelet coefficients from coarse to fine scales, as illustrated in Figure 1. The conditional distribution of each set of wavelet coefficients, given coarse scale coefficients, is sampled with its own (conditional) SGM. The main result is that a normalization of wavelet coefficients allows fixing the same discretization schedule at all scales. Remarkably, and as opposed to existing algorithms, it implies that the total number of sampling iterations per image pixel does not depend on the image size.

After reviewing score-based generation models, Section 2 studies the mathematical properties of its time discretization, with a focus on Gaussian models and multiscale processes. Images and many physical processes are typically non-Gaussian, but do have a singular covariance with long- and short-range correlations. In Section 3, we review how to factorize these processes into probability distributions which capture interactions across scales by introducing orthogonal wavelet transforms. We shall prove that it allows considering SGMs with the same time schedule at all scales, independently of the image size. In Section 4, we present numerical results on Gaussian distributions, the $\varphi^4$ physical model at phase transition, and the CelebA-HQ image dataset [19]. The main contributions of the paper are as follows:

- A Wavelet Score-based Generative Model (WSGM) which generates samples from the conditional distribution of normalized wavelet coefficients, with the same discretization schedule at all scales. The number of time steps per image pixel does not need to depend upon the image size to reach a fixed error level.

- Theorems controlling errors of time discretizations of SGMs, proving accelerations obtained by scale separation with wavelets. These results are empirically verified by showing that WSGM provides an acceleration for the synthesis of physical processes at phase transition and natural image datasets.

## 2 Sampling and Discretization of Score-Based Generative Models

### 2.1 Score-Based Generative Models

**Diffusions and time reversal** A Score-based Generative Model (SGM) [44, 46, 11] progressively maps the distribution of data $x$ into the normal distribution, with a forward Stochastic Differential Equation (SDE) which iteratively adds Gaussian white noise. It is associated with a *noising process* $(x_t)_t$, with $x_0$ distributed according to the data distribution $p$, and satisfying:

$$\mathrm{d}x_t = -x_t \mathrm{d}t + \sqrt{2}\mathrm{d}w_t, \tag{1}$$

where $(w_t)_t$ is a Brownian motion. The solution is an Ornstein-Uhlenbeck process which admits the following representation for any $t \geq 0$:

$$x_t = \mathrm{e}^{-t}x_0 + \sqrt{1 - \mathrm{e}^{-2t}}z, \qquad z \sim \mathcal{N}(0, \mathrm{Id}). \tag{2}$$

The process $(x_t)_t$ is therefore an interpolation between a data sample $x_0$ and Gaussian white noise. The *generative process* inverts (1). Under mild assumptions on $p$ [2, 9], for any $T \geq 0$, the reverse-time process $x_{T-t}$ satisfies:

$$\mathrm{d}x_{T-t} = \{x_{T-t} + 2\nabla \log p_{T-t}(x_{T-t})\} \, \mathrm{d}t + \sqrt{2} \, \mathrm{d}w_t, \tag{3}$$

where $p_t$ is the probability density of $x_t$, and $\nabla \log p_t$ is called the *Stein score*. Since $x_T$ is close to a white Gaussian random variable, one can approximately sample from $x_T$ by sampling from the normal distribution. We can generate $x_0$ from $x_T$ by solving this time-reversed SDE, if we can estimate an accurate approximation of the score $\nabla \log p_t$ at each time $t$, and if we can discretize the SDE without introducing large errors.

Efficient approximations of the Stein scores are the workhorse of SGM. [13] shows that the score $\nabla \log p_t$ can be approximated with parametric functions $s_\theta$ which minimize the so-called implicit

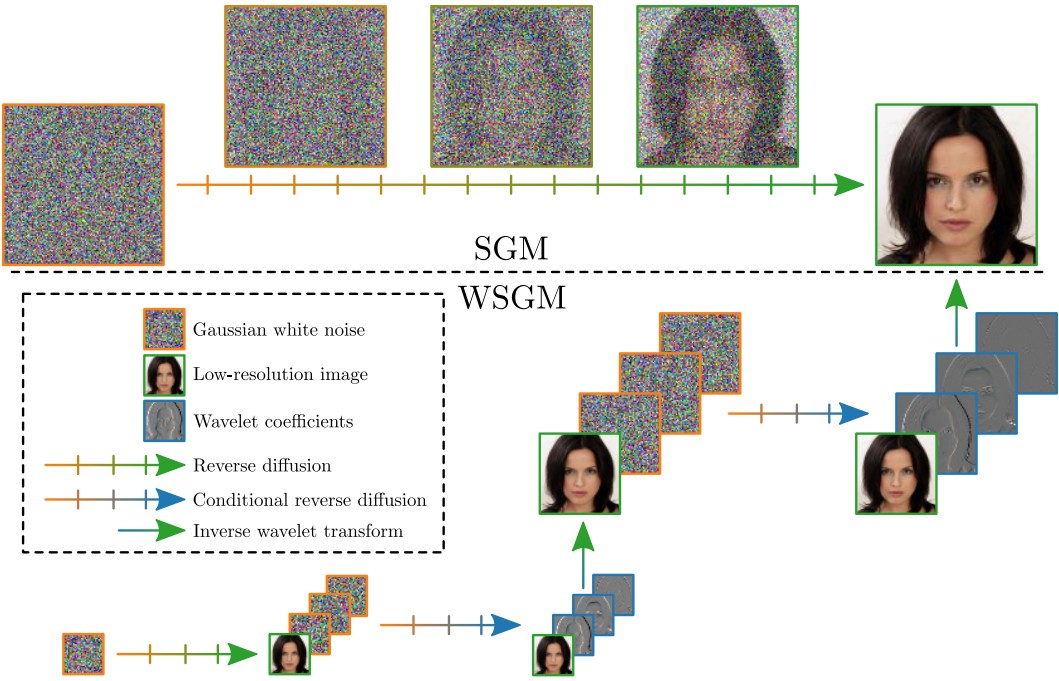

Figure 1: An SGM generates images by discretizing a reverse diffusion, which progressively transforms white Gaussian noise into a natural image. A WSGM generates increasingly higher-resolution images by discretizing reverse diffusions on wavelet coefficients at each scale. It begins by generating a first low-resolution image. Renormalized wavelet coefficients are then generated conditionally to this low-resolution image. A fast inverse wavelet transform reconstructs a higher-resolution image from these wavelet coefficients. This process is repeated at each scale. The number of steps is the same at each scale, and can be orders of magnitude smaller than for SGM.

score matching loss:

$$s_t = \arg\min_\theta \mathbb{E}_{p_t}\left[\frac{1}{2}\|s_\theta(x_t)\|^2 + \operatorname{div}(s_\theta)(x_t)\right], \tag{4}$$

or, equivalently, the denoising score matching loss:

$$s_t = \arg\min_\theta \mathbb{E}_{p_0,\mathcal{N}(0,\mathrm{Id})}\left[\|s_\theta(\mathrm{e}^{-t}x_0 + \sqrt{1-\mathrm{e}^{-2t}}z) + \frac{z}{\sqrt{1-\mathrm{e}^{-2t}}}\|^2\right]. \tag{5}$$

For image generation, $s_\theta$ is calculated by a neural network parameterized by $\theta$. In statistical physics problems where the energy can be linearly expanded with coupling parameters, we obtain linear models $s_\theta(x) = \theta^\top \nabla U(x)$. This is the case for Gaussian processes where $U(x) = xx^\top$; it also applies to non-Gaussian processes, using non-quadratic terms in $U(x)$.

**Time discretization of generation** An approximation of the generative process (3) is computed by approximating $\nabla \log p_t$ by $s_t$ and discretizing time. It amounts to approximating the time-reversed SDE by a Markov chain which is initialised by $\tilde{x}_T \sim \mathcal{N}(0,\mathrm{Id})$, and computed over times $t_k$ which decrease from $t_N = T$ to $t_0 = 0$, at intervals $\delta_k = t_k - t_{k-1}$:

$$\tilde{x}_{t_{k-1}} = \tilde{x}_{t_k} + \delta_k\{\tilde{x}_{t_k} + 2s_{t_k}(\tilde{x}_{t_k})\} + \sqrt{2\delta_k}z_k, \qquad z_k \overset{\text{i.i.d.}}{\sim} \mathcal{N}(0,\mathrm{Id}). \tag{6}$$

Ignoring the error due to the score model, the minimum number of time steps is limited by the Lipschitz regularity of the score $\nabla \log p_t$, see [5, Theorem 1]. The overall complexity of this generation is $N$ evaluations of the score $s_t(x)$.

## 2.2 Discretization of SGM and Score Regularity

We now study how the regularity of the score $\nabla \log p$ affects the discretization of (6). Assuming that the score is known, i.e., that $s_t = \nabla \log p_t$, we prove that for Gaussian processes, the number of time steps to reach a fixed error $\varepsilon$ depends on the condition number of its covariance. This result is generalized to non-Gaussian processes by relating this error to the regularity of $\nabla \log p_t$.

**Gaussian distributions**  Suppose that the data distribution is a Gaussian $p = \mathcal{N}(0, \Sigma)$ with covariance matrix $\Sigma$, in dimension $d$. Let $p_t$ be the distribution of $x_t$. Using (2), we have:

$$\nabla \log p_t(x) = -(\mathrm{Id} + (\Sigma - \mathrm{Id})\mathrm{e}^{-2t})^{-1}x.$$

Let $\tilde{p}_t$ be the distribution of $\tilde{x}_t$ obtained by the time discretization (6). The approximation error between the distribution $\tilde{p}_0$ obtained with the time-reversed SDE and the data distribution $p$ stems from (i) the mismatch between the distributions of $x_T$ and $\tilde{x}_T$, and (ii) the time discretization. The following theorem relates these two errors to the covariance $\Sigma$ of $x$ in the particular case of a uniform time sampling at intervals $\delta_k = \delta$. We normalize the signal energy by imposing that $\mathrm{Tr}(\Sigma) = d$, and we write $\kappa$ the condition number of $\Sigma$, which is the ratio between its largest and smallest eigenvalues.

**Theorem 1.** *If the data distribution $p = \mathcal{N}(0, \Sigma)$, the distribution $\tilde{p}_0$ of $\tilde{x}_0$ in (6) with a uniform discretization $\delta_k = \delta$ satisfies $\mathrm{KL}(p\|\tilde{p}_0) \leq E_T + E_\delta + E_{T,\delta}$, with :*

$$E_T = f(\mathrm{e}^{-4T}|\mathrm{Tr}((\Sigma - \mathrm{Id})\Sigma))|), \tag{7}$$

$$E_\delta = f(\delta|\mathrm{Tr}(\Sigma^{-1} - \Sigma(\Sigma - \mathrm{Id})^{-1}\log(\Sigma)/2 + (\mathrm{Id} - \Sigma^{-1})/3))|), \tag{8}$$

*where $f(t) = t - \log(1 + t)$ and $E_{T,\delta}$ is a higher-order term with $E_{T,\delta} = o(\delta + \mathrm{e}^{-4T})$ when $\delta \to 0$ and $T \to +\infty$. Furthermore, for any $\varepsilon > 0$, there exists $T, \delta \geq 0$ such that:*

$$(1/d)(E_T + E_\delta) \leq \varepsilon \ \text{ and } \ N = T/\delta \leq C\varepsilon^{-2}\kappa^3. \tag{9}$$

*with $C \geq 0$ a universal constant and $\kappa$ the conditioning number of $\Sigma$.*

This theorem specifies the dependence of the Kullback-Leibler error on the covariance matrix. It computes an upper bound on the number of time steps $N = T/\delta$ to reach an error $\varepsilon$ as a function of the condition number $\kappa$ of $\Sigma$. As expected, it indicates that the number of time steps should increase with the condition number of the covariance. This theorem is proved in a more general case in Appendix S5, which includes the case where $p$ has a non-zero mean. An exact expansion of the Kullback-Leibler divergence is also given.

For stationary processes of images, the covariance eigenvalues are given by the power spectrum, which typically decays like $|\omega|^{-1}$ at a frequency $\omega$. It results that $\kappa$ is proportional to a power of the image size. Many physical phenomena produce such stationary images with a power spectrum having a power law decay. In these typical cases, the number of time steps must increase with the image size. This is indeed what is observed in numerical SGM experiments, as seen in Section 3.

**General processes**  Theorem 1 can be extended to non-Gaussian processes. The number of time steps then depends on the regularity of the score $\nabla \log p_t$.

**Theorem 2.** *Assume that $\nabla \log p_t(x)$ is $\mathscr{C}^2$ in both t and x, and that:*

$$\sup_{x,t} \|\nabla^2 \log p_t(x)\| \leq K \ \text{ and } \ \|\partial_t \nabla \log p_t(x)\| \leq M\,\mathrm{e}^{-\alpha t}\|x\|. \tag{10}$$

*for some $K, M, \alpha > 0$. Then $\|p - \tilde{p}_0\|_{\mathrm{TV}} \leq E_T + E_\delta + E_{T,\delta}$, where:*

$$E_T = \sqrt{2}\mathrm{e}^{-T}\,\mathrm{KL}(p\|\mathcal{N}(0, \mathrm{Id}))^{1/2}, \tag{11}$$

$$E_\delta = 6\sqrt{\delta}\,[1 + \mathbb{E}_p(\|x\|^4)^{1/4}]\,[1 + K + M(1 + 1/(2\alpha)^{1/2})], \tag{12}$$

*and $E_{\delta,T}$ is a higher order term with $E_{T,\delta} = o(\sqrt{\delta} + \mathrm{e}^{-T})$ when $\delta \to 0$ and $T \to +\infty$.*

The proof of Theorem 2 is postponed to Appendix S5 and we show that the result can be strengthened by providing a quantitative upper bound on $\|p - \tilde{p}_0\|_{\mathrm{TV}}$. Theorem 2 improves on [5, Theorem 1] by proving explicit bounds exhibiting the dependencies on the regularity constants $K$ and $M$ of the

score and by eliminating an exponential growth term in $T$ in the upper bound. Theorem 2 is much more general but not as tight as Theorem 1.

The first error term (11) is due to the fact that $T$ is chosen to be finite. The second error term (12) controls the error depending upon the discretization time step $\delta$. Since $p_t$ is obtained from $p$ through a high-dimensional convolution with a Gaussian convolution of variance proportional to $t$, the regularity of $\nabla \log p_t(x)$ typically increases with $t$ so $\|\nabla^2 \log p_t(x)\|$ and $\|\partial_t \nabla \log p_t(x)\|$ rather decrease when $t$ increases. This qualitatively explains why a *quadratic* discretization schedule with non-uniform time steps $\delta_k \propto k$ are usually chosen in numerical implementations of SGMs [38, 45]. For simplicity, we focus on the uniform discretization schedule, but our result could be adapted to non-uniform time steps with no major difficulties. This remark also explains that it is mainly the regularity of the score at time $t = 0$ $\nabla \log p$ which determines the error decay (12).

While Theorem 2 is more general than Theorem 1, the Gaussian case provides intuition about the speed of the error decay (12) through the value of the constants $K$ and $M$. If $p$ is Gaussian, then the Hessian $\nabla^2 \log p$ is the negative inverse of the covariance matrix. We verify in Appendix S5 that in this case, the assumptions of Theorem 2 are satisfied. Furthermore, the constants $K$ and $M$, and hence the number of discretization steps, are controlled using the condition number of $\Sigma$. We thus conjecture that non-Gaussian processes with an ill-conditioned covariance matrix will require many discretization steps to have a small error. This will be verified numerically. As we now explain, such processes are ubiquitous in physics and natural image datasets.

**Multiscale processes**   Most images have variations on a wide range of scales. They require to use many time steps to sample using an SGM, because their score is not well-conditioned. This is also true for a wide range of phenomena encountered in physics, biology, or economics [22, 32]. We define a *multiscale process* as a stationary process whose power spectrum has a power law decay. The stationarity implies that its covariance is diagonalized in a Fourier basis. Its eigenvalues, which then coincide with its power spectrum, have a power law decay defined by:

$$P(\omega) \sim (\xi^\eta + |\omega|^\eta)^{-1}, \tag{13}$$

where $\eta > 0$ and $2\pi/\xi$ is the maximum correlation length. Physical processes near phase transitions have such a power-law decay, but it is also the case of many disordered systems such as fluid and gas turbulence. Natural images also typically define stationary processes. Their power spectrum satisfy this property with $\eta = 2$ and $2\pi/\xi \approx L$ for images of size $L \times L$. To efficiently synthesize images and more general multiscale signals, we must eliminate the ill-conditioning properties of the score. This is done by applying a wavelet transform.

## 3   Wavelet Score-Based Generative Model

The numerical complexity of the SGM algorithm depends on the number of time steps, which itself depends upon the regularity of the score. We show that an important acceleration is obtained by factorizing the data distribution into normalized wavelet conditional probability distributions, which are closer to a white Gaussian distribution, and so whose score is better-conditioned.

### 3.1   Wavelet Whitening and Cascaded SGMs

**Normalized orthogonal wavelet coefficients**   Let $x$ be the input signal of width $L$ and dimension $d = L^n$, with $n = 2$ for images. We write $x_j$ its low-frequency approximation subsampled at intervals $2^j$, of size $(2^{-j}L)^n$, with $x_0 = x$. At each scale $2^{j-1} \geq 1$, a fast wavelet orthogonal transform decomposes $x_{j-1}$ into $(\bar{x}_j, x_j)$ where $\bar{x}_j$ are the wavelet coefficient which carries the higher frequency information over $2^n - 1$ signals of size $(2^{-j}L)^n$ [30]. They are calculated with convolutional and subsampling operators $G$ and $\bar{G}$ specified in Appendix S3:

$$x_j = \gamma_j^{-1} G \, x_{j-1} \ \text{ and } \ \bar{x}_j = \gamma_j^{-1} \bar{G} \, x_{j-1} \,. \tag{14}$$

The normalization factor $\gamma_j$ guarantees that $\mathbb{E}[\|\bar{x}_j\|^2] = (2^n - 1)(2^{-j}L)^n$. We consider wavelet orthonormal filters where $(G, \bar{G})$ is a unitary operator, i.e.:

$$\bar{G}G^\top = G\bar{G}^\top = 0 \ \text{ and } \ G^\top G + \bar{G}^\top \bar{G} = \text{Id} \,.$$

It results that $x_{j-1}$ is recovered from $(\bar{x}_j, x_j)$ with:
$$x_{j-1} = \gamma_j\, G^\top x_j + \gamma_j\, \bar{G}^\top \bar{x}_j.$$
The wavelet transform is computed over $J \approx \log_2 L$ scales by iterating $J$ times on (14). The last $x_J$ has a size $(2^{-J}L)^n \approx 1$. Appendix S3 contains a more detailed introduction to the wavelet transform. The choice of wavelet filters $G$ and $\bar{G}$ specifies the properties of the wavelet transform and the number of vanishing moments of the wavelet, as explained in Appendix S4.

**Renormalized probability distribution**    A conditional wavelet renormalization factorizes the distribution $p(x)$ of signals $x$ into conditional probabilities over wavelet coefficients:
$$p(x) = \alpha \prod_{j=1}^{J} \bar{p}_j(\bar{x}_j | x_j)\, p_J(x_J)\,. \tag{15}$$
where $\alpha$ (the Jacobian) depends upon all $\gamma_j$.

Although $p(x)$ is typically highly non-Gaussian, the factorization (15) involves distributions that are closer to Gaussians. The largest scale distribution $p_J$ is usually close to a Gaussian when the image has independent structures, because $x_J$ is an averaging of $x$ over large domains of size $2^J$. In images, the wavelet coefficients $\bar{x}_j$ are usually sparse and thus have a highly non-Gaussian distribution; however, it has been observed [48] that their conditional distributions $\bar{p}_j(\bar{x}_j | x_j)$ become much more Gaussian, due to dependencies of wavelet coefficients across scales. Furthermore, because of the renormalization, the normalized wavelet coefficients $\bar{x}_j$ have a white spectrum, as opposed to a power-law decay for $x_j$, which implies they are closer to a white Gaussian distribution. In statistical physics, the analysis of high frequencies conditioned by lower frequencies have been studied in [50]. More recently, normalized wavelet factorizations (15) have been introduced in physics to implement renormalization group calculations, and model probability distributions with maximum likelihood estimators near phase transitions [33].

**Wavelet Score-based Generative Model**    Instead of computing a Score-based Generative Model (SGM) of the distribution $p(x)$, a Wavelet Score-based Generative Model (WSGM) applies an SGM at the coarsest scale $p_J(x_J)$ and then on each conditional distribution $\bar{p}_j(\bar{x}_j | x_j)$ for $j \le J$. It is thus a cascaded SGM, similarly to [12, 40], but calculated on $\bar{p}_j(\bar{x}_j | x_j)$ instead of $p_j(x_{j-1} | x_j)$. The normalization of wavelet coefficients $\bar{x}_j$ effectively produces a whitening which can considerably accelerate the algorithm by reducing the number of time steps. This is not possible on $x_{j-1}$ because its covariance is ill-conditioned. It will be proved for Gaussian processes.

A forward noising process is computed on each $\bar{x}_j$ for $j \le J$ and $x_J$:
$$\mathrm{d}\bar{x}_{j,t} = -\bar{x}_{j,t}\,\mathrm{d}t + \sqrt{2}\mathrm{d}\bar{w}_{j,t} \ \text{ and } \ \mathrm{d}x_{J,t} = -x_{J,t}\,\mathrm{d}t + \sqrt{2}\mathrm{d}w_{J,t},$$
where the $\bar{w}_{j,t}, w_{J,t}$ are Brownian motions. Since $\bar{x}_j$ is nearly white and has Gaussian properties, this diffusion converges much more quickly than if applied directly on $x$. Using (4) or (5), we compute a score function $s_{J,t}(x_{J,t})$ which approximates the score $\nabla \log p_{J,t}(x_{J,t})$. For each $j \le J$ we also compute the conditional score $\bar{s}_{j,t}(\bar{x}_{j,t} | x_j)$ which approximates $\nabla \log \bar{p}_{j,t}(\bar{x}_{j,t} | x_j)$.

The inverse generative process is computed from coarse to fine scales as follows. At the largest scale $2^J$, we sample the low-dimensional $x_J$ by discretizing the inverse SDE. Similarly to (6), the generative process is given by:
$$x_{J,t_{k+1}} = x_{J,t_k} + \delta_k\{x_{J,t_k} + 2s_{J,t_k}(x_{J,t_k})\} + \sqrt{2\delta_k}z_{J,k}, \qquad z_{J,k} \overset{\text{i.i.d.}}{\sim} \mathcal{N}(0, \mathrm{Id}). \tag{16}$$
For $j$ going from $J$ to 1, we then generate the wavelet coefficients $\bar{x}_j$ conditionally to the previously calculated $x_j$, by keeping the same time discretization schedule at all scales:
$$\bar{x}_{j,t_{k+1}} = \bar{x}_{j,t_k} + \delta_k\{\bar{x}_{j,t_k} + 2\bar{s}_{j,t_k}(\bar{x}_{j,t_k} | x_j)\} + \sqrt{2\delta_k}\,z_{j,k}, \qquad z_{j,k} \overset{\text{i.i.d.}}{\sim} \mathcal{N}(0, \mathrm{Id}). \tag{17}$$
The inverse wavelet transform then approximately computes a sample of $x_{j-1}$ from $(\bar{x}_{j,0}, x_j)$:
$$\tilde{x}_{j-1} = \gamma_j\, G^\top x_j + \gamma_j\, \bar{G}^\top \bar{x}_{j,0}. \tag{18}$$
The generative process is illustrated in Figure 1 and its pseudocode is given in Algorithm 1 in Appendix S2. The appendix also verifies that if $x$ is of size $d$ then the numerical complexity of the generation is $O(Nd)$, where $N$ is the number of time steps, which is the same at each scale. For multiscale processes, we shall see that the number of time steps $N$ does not depend upon $d$ to reach a fixed error measured with a KL divergence.

**Related work** Multi-scale representations, based on wavelets or not, have been incorporated in many generative modeling approaches in order to increase generation quality and sampling efficiency. Specifically, they have been shown to improve results for auto-encoders [4], GANs [7] and normalizing flows [26]. Closer in spirit to our work, [52] introduces Wavelet Flow, a normalizing flow with a cascade of layers generating wavelet coefficients conditionally on lower-scales, then aggregating them with an inverse wavelet transform. This method yields training time acceleration and high-resolution ($1024 \times 1024$) generation.

WSGM is closely related to other cascading diffusion algorithms, such as the ones introduced in [12, 40, 6]. The main difference lies in that earlier works on cascaded SGMs do not model the *wavelet coefficients* $\{\bar{x}_j\}_{j=1}^J$ but the *low-frequency* coefficients $\{x_j\}_{j=1}^J$. As a result, cascaded models do not explicitly exploit the whitening properties of the wavelet transform, nor the fact that conditional wavelet distributions are often nearly Gaussian, and the mechanisms behind the acceleration remain implicit. We also point out the recent work of [16] which, while not using the cascading framework, drop subspaces from the noising process at different times. This allows using only one SDE to sample approximately from the data distribution. However, the reconstruction is still computed with respect to $\{x_j\}_{j=1}^J$ instead of the wavelet coefficients.

Finally, we highlight that our work could be combined with other acceleration techniques such as the ones of [17, 27, 53, 42, 37, 43, 11, 23, 29, 41, 51] in order to improve the empirical results of WSGM.

## 3.2 Discretization and Accuracy for Gaussian Processes

We now illustrate Theorem 1 and the effectiveness of WSGM on Gaussian multiscale processes. We use the whitening properties of the wavelet transform to show that the time complexity required in order to reach a given error is linear in the image dimension.

The following result proves that the normalization of wavelet coefficients performs a preconditioning of the covariance, whose eigenvalues then remain of the order of 1. This is a consequence of a theorem proved by [34] on the representation of classes of singular operators in wavelet bases, see Appendix S4. As a result, the number of iterations $N = T/\delta$ required to reach an error $\varepsilon$ is independent of the dimension.

**Theorem 3.** *Let $x$ be a Gaussian stationary process of power spectrum $P(\omega) = c\,(\xi^\eta + |\omega|^\eta)^{-1}$ with $\eta > 0$ and $\xi > 0$. If the wavelet has a compact support, $q \geq \eta$ vanishing moments and is $\mathscr{C}^q$, then the first-order terms $E_T$ and $E_\delta$ in the sampling error of WSGM $\mathrm{KL}(p\|\tilde{p}_0)$ are such that for any $\varepsilon > 0$, there exists $C > 0$ such that for any $\delta$, $T$:*

$$(1/d)(E_T + E_\delta) \leq \varepsilon \ \text{ and } \ N = T/\delta \leq C\varepsilon^{-2}. \tag{19}$$

To prove this result, we show that the conditioning number of the covariance matrix of the renormalized wavelet coefficients does not depend on the dimension, by using Sobolev norm equivalences [15, 34]. We conclude upon combining this result, the cascading property of the Kullback-Leibler divergence and an extension of Theorem 1 to the setting with non-zero mean. The detailed proof is postponed to Appendix S6.

**Numerical results** We illustrate Theorem 3 on a Gaussian field $x$, whose power spectrum $P$ has a power law decay (13). In Figure 2, we display the sup-norm between $P$ and the power spectrum $\hat{P}$ of the samples obtained using either vanilla SGM or WSGM with uniform stepsize $\delta_k = \delta$. In the case of vanilla SGM, the number $N(\varepsilon)$ of time steps needed to reach a small error $\|P - \hat{P}\| = \varepsilon$ increases with the size of the image $L$ (Fig. 2, right). Equation (9) suggests that $N(\varepsilon)$ scales like a power of the conditioning number $\kappa$ of $\Sigma$, which is for multiscale Gaussian processes $\kappa \sim L^\eta$, for images of size $L \times L$. In the WSGM case, we sample from the conditional distributions $\bar{p}_j$ of wavelet coefficients $\bar{x}_j$ given low frequencies $x_j$. At a scale $j$, the conditioning numbers $\bar{\kappa}_j$ of the conditional covariance become dimension-independent (Appendix S4), removing the dependency of $N(\varepsilon)$ on the image size $L$ as suggested by (19).

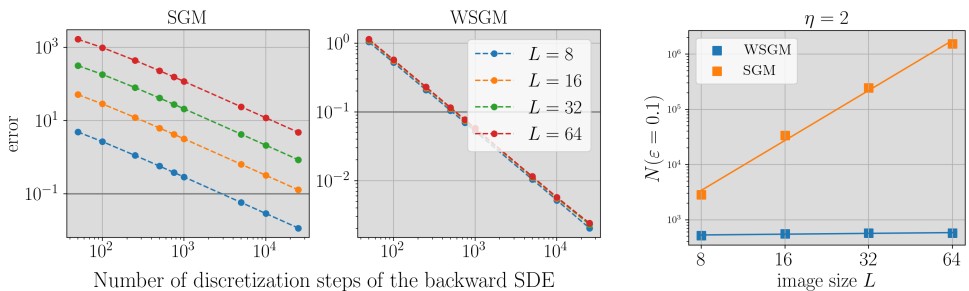

Figure 2: **Left and middle:** evolution of the error on the estimated covariance matrix using either SGM or WSGM w.r.t. the number of stepsizes used in the model ($T = 10$ is fixed). **Right:** number $N(\varepsilon)$ of discretization steps required to reach a given error $\varepsilon = 0.1$ using either SGM or WSGM.

## 4 Acceleration with WSGM: Numerical Results

For multiscale Gaussian processes, we proved that with WSGMs, the number of time steps $N(\varepsilon)$ to reach a fixed error $\varepsilon$ does not depend on the signal size, as opposed to SGMs. This section shows that this result applies to non-Gaussian multiscale processes. We consider a physical process near a phase transition and images from the CelebA-HQ database [19].

### 4.1 Physical Processes with Scalar Potentials

Gaussian stationary processes are maximum entropy processes conditioned by second order moments defined by a circulant matrix. More complex physical processes are modeled by imposing a constraint on their marginal distribution, with a so-called scalar potential. The marginal distribution of $x$ is the probability distribution of an image pixel $x(u)$, which does not depend upon $u$ if $x$ is stationary. Maximum entropy processes conditioned by second order moments and marginal distributions have a probability density which is a Gibbs distribution $p(x) = Z^{-1} \, e^{-E(x)}$ with:

$$E(x) = \tfrac{1}{2}x^\top C x + \sum_u V(x(u)) \ , \tag{20}$$

where $C$ is a circulant matrix and $V \colon \mathbb{R} \to \mathbb{R}$ is a scalar potential. Appendix S8 explains how to parameterize $V$ as a linear combination of a family of fixed elementary functions. The $\varphi^4$ model is a particular example where $C = -\Delta$ is the negative Laplacian and $V$ is a fourth-order polynomial, adjusted in order to impose that $x(u) \approx \pm 1$ with high probability. For so-called critical values of these parameters, the resulting process becomes multiscale with long range interactions and a power law spectrum, see Figure 3-(c).

We train SGMs and WSGMs on critical $\varphi^4$ processes of different sizes; for the score model $s_\theta$, we use a simple linear parameterization detailed in Appendix S8.2. To evaluate the quality of the generated samples, it is sufficient to verify that these samples have the same second order moment and marginals as $\varphi^4$. We define the error metric as the sum of the L$^2$ error on the power spectrum and the total-variation distance between marginal distributions. Figure 3-(a) shows the decay of this error as a function of the number of time steps used in an SGM and WSGM with a uniform discretization. With vanilla SGM, the loss has a strong dependency in $L$, but becomes almost independent of $L$ for WSGM. This empirically verifies the claim that an ill-conditioned covariance matrix leads to slow sampling of SGM, and that WSGM is unaffected by this issue by working with the conditional distributions of normalized wavelet coefficients.

### 4.2 Scale-Wise Time Reduction in Natural Images

Images are highly non-Gaussian multiscale processes whose power spectrum has a power law decay. We now show that WSGM also provides an acceleration over SGM in this case, by being independent of the image size.

We focus on the CelebA-HQ dataset [28] at the $128 \times 128$ resolution. Its power spectrum has a power law decay, as shown in Figure 4, and it thus suffers from ill-conditioning, even though it is a

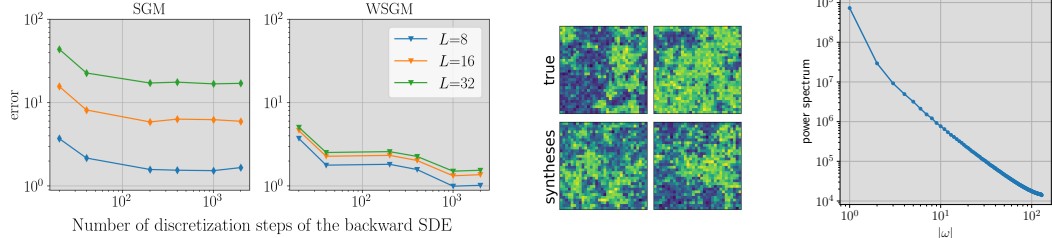

Figure 3: **Left:** error between ground-truth $\varphi^4$ datasets in various dimensions $L$, and the synthetized datasets with SGM and WSGM, for various number of discretization steps. **Middle:** realizations of $\varphi^4$ (top) and WSGM samples (bottom). **Right:** power spectrum of $\varphi^4$ for $L = 256$.

non-stationary process. We compare SGM [11] samples at the $128 \times 128$ resolution with WSGM samples which start from the $32 \times 32$ resolution. Though smaller, the $32 \times 32$ resolution still suffers from a power law decay of its spectrum over several orders of magnitude. The reason why we limit this coarsest resolution is because border effects become dominant at lower image sizes. To simplify the handling of border conditions, we use Haar wavelets.

Following [38], the global scores $s_\theta(x)$ are parameterized by a neural network with a U-Net architecture. It has 3 residual blocks at each scale, and includes multi-head attention layers at lower scales. The conditional scores $s_\theta(\bar{x}_j | x_j)$ are parameterized in the same way, and the conditioning on the low frequencies $x_j$ is done with a simple input concatenation along channels [38, 40]. The details of the architecture are in Appendix S9. We use a uniform discretization of the backward SDE to stay in the setting of Theorem 2, and show that WSGM still obtains satisfactory results in this case.

The generation results are given in Figure 4. With the same computational budget of 16 discretizations steps at the largest scale (iterations at smaller scales having a negligible cost due to the exponential decrease in image size), WSGM achieves a much better perceptual generation quality. Notably, SGM generates noisy images due to discretization errors. This is confirmed quantitatively with the Fréchet Inception Distance (FID) [10]. The FID of the WSGM generations decreases with the number of steps, until it plateaus. This plateau is reached with at least 2 orders of magnitude less steps for WSGM than SGM. This number of steps is also independent of the image size for WSGM, thus confirming the intuition given in the Gaussian case by Theorems 1 and 3. Our results confirm that vanilla SGM on a wide range of multiscale processes, including natural images, suffers from ill-conditioning, in the sense that the number of discretization steps grows with the image size. WSGM, on the contrary, leads to uniform discretization schemes whose number of steps at each scale does not depend on the image size.

We also stress that there exists many techniques [18, 17, 27, 53, 42, 37, 43, 23, 11, 29, 41, 51] to accelerate the sampling of vanilla SGMs, with sometimes better FID-time complexity tradeoff curves. Notably, the FID plateaus at a relatively high value of 20 because the coarsest resolution $32 \times 32$ is still ill-conditioned, and thus requires thousands of steps with a non-uniform discretization schedule to achieve FIDs less than 10 with vanilla SGM [38]. Such improvements (including proper handling of border conditions) are beyond of the scope of this paper. The contribution of WSGM is rather to show the reason behind this sampling inefficiency and mathematically prove in the Gaussian setting that wavelet decompositions of the probability distribution allows solving this problem. Extending this theoretical result to a wider class of non-Gaussian multiscale processes, and combining WSGM with other sampling accelerations, are interesting research directions.

## 5   Discussion

This paper introduces a Wavelet Score-based Generative Model (WSGM) which applies an SGM to normalized wavelet coefficients conditioned by lower frequencies. We prove that the number of steps in SGMs is controlled by the regularity of the score of the target distribution. For multiscale processes such as images, it requires a considerable number of time steps to achieve a good accuracy, which increases quickly with the image size. We show that a WSGM eliminates ill-conditioning issues by normalizing wavelet coefficients. As a result, the number of steps in WSGM does not increase with

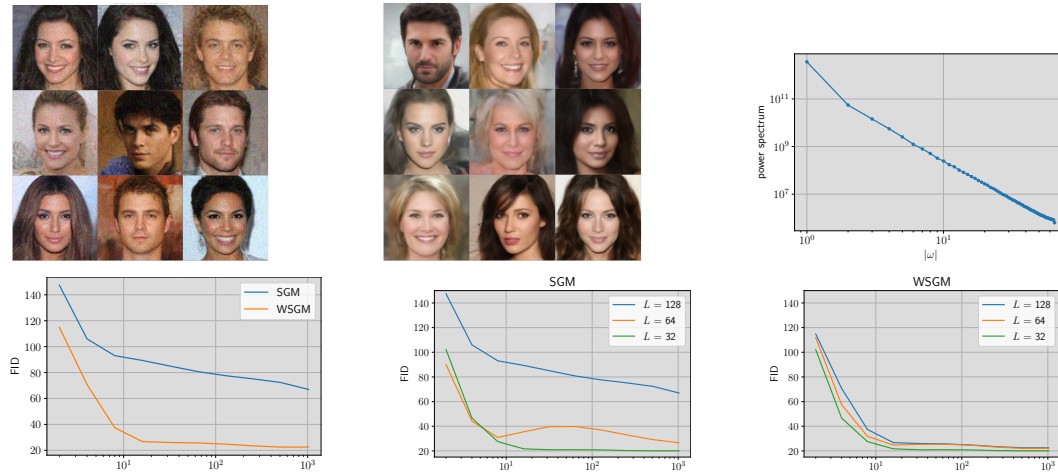

Figure 4: **Top.** (a): Generations from SGM with 16 discretization steps. (b): Generations from WSGM with 16 discretization steps at each scale. (c): Power spectrum of CelebA-HQ. **Bottom.** (a): Evolution of the FID w.r.t. the number of diffusion steps for SGM and WSGM with $L = 128$. (b): Evolution of the FID w.r.t. the number of diffusion steps for SGM at several image sizes $L$. (c) Evolution of the FID w.r.t. the number of diffusion steps for WSGM at several image sizes $L$.

the image size. We illustrated our results on Gaussian distributions, physical processes and image datasets.

One of the main limitations of WSGM is that it is limited to multiscale processes for which the conditional wavelet probabilities are nearly white. A promising direction for future work is to combine WSGM with other acceleration techniques such as adaptive time discretizations to handle such cases. In another direction, one could strengthen the theoretical study of SGM and extend our results beyond the Gaussian setting, in order to fully describe SGM on physical processes that can be seen as perturbations of Gaussian distributions.

### Acknowledgments

This work was supported by a grant from the PRAIRIE 3IA Institute of the French ANR-19-P3IA-0001 program. We would like to thank the Scientific Computing Core at the Flatiron Institute for the use of their computing resources.

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
