# OpenReview forum: "Wavelet Score-Based Generative Modeling"
_NeurIPS.cc/2022/Conference — NeurIPS 2022 Accept_

### Official Review · Reviewer_6XjU · 2022-07-09

**Rating:** 7
**Confidence:** 3
**Soundness:** 4 excellent
**Presentation:** 3 good
**Contribution:** 4 excellent

**Summary:**

In this paper, the authors attempted to alleviate the high computation cost in Score-based generative models (SGMs) by reducing the number of time steps. In particular, the authors first found that the number of times steps in SGMs to reach a fixed error depends on the condition number of the Gaussian covariance. To break this dependence, they proposed a novel SGMs with wavelet whitening. Solid theoretical and technical contributions have been witnessed in this paper.

**Questions:**

Is the proposed model competitive compared with other baselines to accelerate SGMs, such as the models in referred paper "[16, 15, 23, 46, 36, 31, 37, 20, 10, 25, 35, 45] "?

**Strengths And Weaknesses:**

Strengths:

1.The authors provide a theoretical analysis of how the regularity of the stein score affects the discretization of generation. They found that, to reach the fixed error, the number of time steps increases with the condition number of the covariance.

2.It is an interesting and novel idea to utilize the wavelet orthogonal transform to decompose and subsample the input image, which can reduce the image size.

3.The authors prove that the proposed model can fix the same discretization schedule for all image scales and accelerate the generation in SGMs, where the time complexity grows linearly with the image size.

Weakness:

1.The authors should provide more descriptions of the wavelet transforms in this paper. It is hard for me to understand the major idea in this paper before learning some necessary knowledge about wavelet whitening, wavelet coefficient, and so on.

2.It is better for authors to display the performance of accelerating SGMs by involving some other baselines with a different perspective, such as “optimizing the discretization schedule or by modifying the original SGM formulation” [16, 15, 23, 46, 36, 31, 37, 20, 10, 25, 35, 45]

---

> ### Author Response · Authors · 2022-08-02
> **Official Answer to Reviewer 6XjU**
>
> We thank the reviewer for their time and effort spent on their review.
>
> The reviewer rightfully points out that the paper lacks background on the wavelet transform. In addition to the appendix S3 (now S4) on the links between the discrete and continuous orthogonal wavelet bases, we also added an appendix (new appendix S3) to provide a non-technical introduction to wavelet transforms to help make the paper more accessible.
>
> The reviewer then asks about other acceleration baselines. We stress that the acceleration provided by WSGM comes from increasing the regularity of the score, which is orthogonal to and therefore should combine with most other acceleration techniques. We chose to stick to e.g. uniform discretization to stay in the setting of the Theorems in the paper. However, we are currently running some of our experiments with DDIM and non-uniform discretization to combine our acceleration with the one from [1].
>
> [1] Song et al. (2021) – Denoising Diffusion Implicit Models

---

> > ### Comment · Reviewer_6XjU · 2022-08-05
> > **Reviewer 6XjU**
> >
> > Thanks for your responses. Overall, I stick to the score 7.

---

### Official Review · Reviewer_4Ru5 · 2022-07-10

**Rating:** 6
**Confidence:** 3
**Soundness:** 3 good
**Presentation:** 4 excellent
**Contribution:** 3 good

**Summary:**

This paper proposes Wavelet score-based generative model (WSGM), a multi-scale score-based generative model with an accelerated sampling by reducing the number of required iterations of the score estimator leveraged by renormalized wavelet coefficients. An important property of WSGM is a theoretical analysis on the required number of sampling iterations to reach the desired error which is not dependent on the dimension of the signal per scale.

**Questions:**

Understanding that the high-resolution image experiments may be challenging during the discussion period, do the authors have an opinion on expected outcome? Are there practical blockers of WSGM for the scaled-up evaluation? I believe that a high-impact work related to the diffusion based models often comes with the promising empirical studies using real-world high-resolution datasets.

Accelerating SGM inference is one of the most actively studied area, which naturally deviates from the controlled experimental setup of WSGM (uniform discretization). Can the authors consider more realistic experimental setups using several popular acceleration methods? What will be the implication from such assessment?

**Limitations:**

The authors have addressed the limitation of WSGM, that is, using uniform discretization schedule for SGM sampling which does not represent modern SGMs.

**Strengths And Weaknesses:**

### Strengths
The paper presents a rigorous theoretical proof and mathematical analysis on the connection between the normalized wavelet coefficients and the sampling iterations of SGM. The proposed method, the renormalized wavelet decompositions, provides a principled approach to reduce the Lipschitz irregularity [De Bortoli+2021] from the score estimator.

The experimental results on three classes (Gaussian process, physical process, and natural images) support the theoretical foundation of WSGM, where it surpasses the baseline multi-scale SGM under the same number of discretized reverse diffusion steps. Although the considered baseline model is not based on recent state-of-the-art SGM (uniform discretization for sampling, for example), the controlled experiments in a principled way are convincing enough for me to justify the rigorous theoretical contribution of WSGM.

### Weaknesses
Although the wavelet decomposition-based approach is well motivated with mathematical connections, I would like to point out that such approach for the multi-scale generative model has also  been studied, such as Wavelet Flow [Yu+2020] on normalizing flows, or SWAGAN [Gal+2021], for example. Although the main scope of this paper is SGM, I suggest allocating paragraphs for a description of related work using wavelet approach for the multi-scale generative model, to present a complete picture on the considered approach.

The current scope of empirical experiments is sound, but I consider it a missed opportunity to show the result on WSGM on the high-resolution images. Considering the recent progress on SGM on high-fidelity synthesis of images, WSGM is expected to have a potential to achieve a high-resolution image modeling (1024x1024 for example) which is currently not easily approachable. In fact, [Yu+2020] showcased their method as the first to unlock model training and sampling on 1024x1024 images. I believe that additional experiments on the high-resolution images will strengthen the impression of WSGM also from the practical point of view.

---

> ### Author Response · Authors · 2022-08-02
> **Official Answer to Reviewer 4Ru5**
>
> We thank the reviewer for their time and effort spent on their review.
>
> We agree with the reviewer that a broader overview of related work on multiscale generation would provide additional context to the paper. We added a paragraph to this end (lines 211 - 217).
>
> We also agree with the reviewer that the paper would benefit from a more thorough evaluation of WSGM in more large-scale settings. The main issue is computational in nature, as larger image resolutions require larger score networks. We chose not to include such larger-scale experiments as the focus of the paper is on the sampling rather than the training of the score networks. We however expect that WSGM would keep its advantage over SGM in those cases. We are currently running additional experiments on 1024x1024 images (CelebA-HQ) and will share them with the reviewer as soon as we get the experimental results (we highlight that such experiments are time-consuming given our hardware limitations).
>
> The reviewer then asks about more realistic experimental setups with acceleration methods. We stress that the acceleration provided by WSGM comes from increasing the regularity of the score, which is orthogonal to and therefore should combine with most other acceleration techniques (like improved samplers [1,2,3], knowledge distillation [4] or DDIM [5]) . In our experiments we chose to stick to e.g. uniform discretization to stay in the setting of the Theorems in the paper. We are also currently running some of our experiments with DDIM and non-uniform discretization to combine our acceleration with the one from [5].
>
> [1] Jolicoeur-Martineau et al. (2022) – Gotta go fast when generating data with score-based models
>
> [2] Liu et al. (2022) – Pseudo numerical methods for diffusion models on manifolds
>
> [3] Zhang et al. (2022) – Fast Sampling of Diffusion Models with Exponential Integrator
>
> [4] Salimans et al. (2022) – Progressive distillation for fast sampling of diffusion models
>
> [5] Song et al. (2021) – Denoising Diffusion Implicit Models

---

> > ### Comment · Reviewer_4Ru5 · 2022-08-08
> > **Thanks**
> >
> > Thank you for the rebuttal. I acknowledge your rebuttal and the other reviewers.
> > I am keeping my score as 6 during the author-reviewer rebuttal, because I still consider it the missed opportunity to see the high-resolution image experiments at submission.
> > Still, I am supportive on this work, and I would like to see the authors to include the aforementioned experimental results to the final version of the paper. I think this will strengthen this work significantly.

---

### Official Review · Reviewer_FAib · 2022-07-11

**Rating:** 7
**Confidence:** 3
**Soundness:** 4 excellent
**Presentation:** 4 excellent
**Contribution:** 4 excellent

**Summary:**

This paper introduces the wavelet diffusion model, which is theoretically guaranteed to achieve $\epsilon$ error, independent of the step size.

**Questions:**

-

**Limitations:**

-

**Strengths And Weaknesses:**

$\textbf{Strengths}$
- This work provides theoretically and empirically sufficient evidence that the wavelet diffusion model has a fixed number of function evaluations to synthesize an image, independently of the resolution.
- In particular, the theory is excellent, and the arguments are solid.

$\textbf{Weaknesses}$
- In contrast to the theory, the experiments are disappointing. The authors only demonstrate their claim on the CelebA-HQ dataset, which significantly differs from the ImageNet or CIFAR-10 dataset in its diversity. If the suggested wavelet diffusion model is truly effective, then at least the authors should have proved its efficacy on various datasets.
- The number of sampling iterations to reach an error $\epsilon$ is independent of the dimension only for the case of the Gaussian process. I'm a fan of this paper, but the lack of the analogous proof of Theorem 2 to the wavelet diffusion model is a bit unsatisfactory, though I understand how difficult it is to prove.
- Though the paper has a clear and solid lesson, it is still largely missing how to use this lesson in practice. For instance, it would be much better to compare the vanilla SGM and the WSGM when the number of discretization increases to $O(1000)$. I suspect that SGM outperforms WSGM as $L>1000$, is it right? If then, the authors should clearly point out that as their limitations.

---

> ### Author Response · Authors · 2022-08-02
> **Official Answer to Reviewer FAib**
>
> We thank the reviewer for their time and effort spent on their review.
>
> The reviewer rightfully points out that WSGM is only evaluated on the CelebA-HQ dataset for natural images. The main issue is computational in nature, as more diverse datasets require larger score networks. We chose not to include such larger-scale experiments as the focus of the paper is on the sampling rather than the training of the score networks. We however expect that WSGM would keep its advantage over SGM in those cases. Though the rebuttal period is too short to add new results, we will try to include results on the ImageNet dataset for the camera-ready version. CIFAR-10 is however unsuitable because of its too small 32x32 resolution to see the improvements of WSGM.
>
> We agree with the reviewer that the paper would benefit from the equivalent of Theorem 2 for WSGM. Theorem 2 can be applied to the intermediate diffusions at each scale in WSGM. The error is then bounded by the Lipschitz regularity of the conditional scores instead of the global score. The issue is to obtain non-trivial bounds on these conditional scores, which requires assumptions on the data distribution. Designing good models of non-Gaussian image processes which allow obtaining such bounds is an interesting avenue for future work.
>
> Finally, the reviewer asks about the comparison between SGM and WSGM for large numbers of steps. We want to point out that the experiments in the paper go up to a thousand discretization steps. Figure 4 shows that even with 1000 discretization steps, WSGM significantly outperforms SGM in terms of FID on CelebA-HQ.

---

> > ### Comment · Reviewer_FAib · 2022-08-04
> > **Response to Authors**
> >
> > (Dataset) There is no systematic analysis of the sampling and the data diversity, and I think that a data with diversity would behave differently in sampling procedure. However, the paper's contribution is significant, and I would not make it an issue. An additional result is, still, highly recommendable.
> >
> > (Number of Steps) It seems that I missed Figure 4. Figure 4 successfully solves my concern.
> >
> > Overall, I stick to the score 7 for considering WSGM's contribution. This is a very good paper, and I hope to see this paper in arXiv as soon as possible.

---

### Meta-Review · Area_Chair_Kv2p · 2022-08-23

**Recommendation:** Accept
**Confidence:** Certain

**Metareview:**

Summary: This paper introduce Wavelet Score-Based Generative Modeling, a multi-scale diffusion models that allows for considerably faster synthesis by working in the wavelet basis. The topic is timely, given the community's growing interest in diffusion models, and the high computational cost of sampling from such models. The reviewers think that this is a technically solid paper that introduces important insight, for example that the number of required time steps increases with the condition number. The reviewers found the empirical comparison to a baseline diffusion model satisfactory The reviewers had some relatively minor concerns, some of were addressed in a healthy discussion. For example. the submitted paper lacked an introductory explanation of the wavelet transform, but this was addressed by the authors by including an additional appendix. The reviewers agree that the theoretical and empirical contributions are sufficient for publication. The reviewers do note that experiments on larger-scale and/or more diverse datasets (e.g 1024x1024) would be illuminating. There was no discussion among reviewers.

Recommendation: I recommend to accept this paper.

**Award:**

No

---

### Decision · Program_Chairs · 2022-09-14

Accept